# Recent Advances in Analytical Methods for Determination of Polyphenols in Tea: A Comprehensive Review

**DOI:** 10.3390/foods11101425

**Published:** 2022-05-13

**Authors:** Mu-Fang Sun, Chang-Ling Jiang, Ya-Shuai Kong, Jin-Lei Luo, Peng Yin, Gui-Yi Guo

**Affiliations:** 1Henan Engineering Research Center of Tea Processing and Testing, Henan Key Laboratory of Tea Plant Comprehensive Utilization in South Henan, College of Tea Science, Xinyang Agriculture and Forestry University, Xinyang 464000, China; xynzsmf@163.com (M.-F.S.); 2021360002@xyafu.edu.cn (C.-L.J.); 2021360001@xyafu.edu.cn (Y.-S.K.); 2020360005@xyafu.edu.cn (J.-L.L.); ggy6363@aliyun.com (G.-Y.G.); 2National Research Center of Engineering and Technology for Utilization of Botanical Functional Ingredients, Co-Innovation Center of Education Ministry for Utilization of Botanical Functional Ingredients, Key Laboratory of Tea Science of Ministry of Education, Hunan Agricultural University, Changsha 410128, China

**Keywords:** polyphenols, tea, analytical method, liquid chromatography (LC), high resolution mass spectrometry (HRMS)

## Abstract

Polyphenols, the most abundant components in tea, determine the quality and health function of tea. The analysis of polyphenols in tea is a topic of increasing interest. However, the complexity of the tea matrix, the wide variety of teas, and the difference in determination purposes puts forward higher requirements for the detection of tea polyphenols. Many efforts have been made to provide a highly sensitive and selective analytical method for the determination and characterization of tea polyphenols. In order to provide new insight for the further development of polyphenols in tea, in the present review we summarize the recent literature for the detection of tea polyphenols from the perspectives of determining total polyphenols and individual polyphenols in tea. There are a variety of methods for the analysis of total tea polyphenols, which range from the traditional titration method, to the widely used spectrophotometry based on the color reaction of Folin–Ciocalteu, and then to the current electrochemical sensor for rapid on-site detection. Additionally, the application of improved liquid chromatography (LC) and high-resolution mass spectrometry (HRMS) were emphasized for the simultaneous determination of multiple polyphenols and the identification of novel polyphenols. Finally, a brief outline of future development trends are discussed.

## 1. Introduction

Tea is one of the most popular beverages in the world, second only to water, and far ahead of carbonated soft drinks, wine, beer, and coffee [1]. The tea plant (*Camellia sinensis* (L.) O. Kuntze) first originated in southwest China and has a history of about 5000 years. So far, more than 60 countries plant tea crop, and people in more than 160 countries have the habit of drinking tea [2]. In addition, based on the business data platform Statista, the global tea beverage market is estimated at approximately USD 44.3 billion in 2021 [3]. Tea is mainly processed from the young tea shoots. Fresh tea leaves contain high contents of water, rich astringent, bitter ingredients, and a faint aroma. However, various flavor compounds are formed after processing. The differences in processing technology result in six major tea categories with different degrees of fermentation: green tea, white tea, yellow tea, oolong tea, black tea, dark tea (Figure 1). Among these tea categories, black tea is the most consumed, accounting for about 75% of global tea consumption, and five times more than that of green tea [4].

The acceptance and preference of tea by people from different countries and nationalities are largely related to its beneficial effects on health. Tea has been regarded as a traditional Chinese medicine that can improve or prevent various diseases since ancient times. Tea is rich in polyphenols, the content of polyphenols in fresh tea leaves is generally 18% to 36% (dry weight), with strong anti-inflammatory, antioxidant, and anti-mutation physiological characteristics. Many animal and clinical studies have shown that tea polyphenols play a vital role in the prevention of cardiovascular diseases, cancer, obesity, diabetes, and allergic diseases [5,6,7,8,9,10]. Besides, as the main secondary metabolites in tea, tea polyphenols are often regarded as physiological markers in identifying tea quality [11], distinguishing tea plant varieties [12], raw material tenderness [13], analyzing processing reaction mechanisms [14], judging tea storage time [15], and so forth. It is of great significance to the promotion of tea standardization and the development of the tea industry.

Accurate analysis technology is crucial to deeply analyse the physiological activity of tea polyphenols and its development and utilization. However, tea contains a lot of polyphenolic compounds, and the content varies greatly (the concentration unit from μg/kg to mg/g) [14,16]. In addition to some abundant tea polyphenols such as catechins and flavonoids, there are also many low-content and unidentified polyphenols in tea. Furthermore, different tea plant varieties, different processed teas, and different matrices (fresh leaves, dry teas, and tea infusions) make the determination of tea polyphenols difficult.

Spectrophotometry and high-performance liquid chromatography (HPLC) are often used for the determination of polyphenols in tea, and these traditional analytical methods have been used as the international standard methods (ISO 14502-1:2005 and ISO 14502-2:2005) and China’s national standard methods (GB/T 8313-2018) for the determination of tea polyphenols, and laid a solid foundation for the development of tea science. However, with the deepening of scientific research on tea, researchers have put forward higher requirements for the accurate quantitative analysis of functional components in tea. In recent years, with the rapid development of analytical technology, especially tandem mass spectrometry and high-resolution mass spectrometry (HRMS), analytical instruments have made rapid progress in improving sensitivity, precision, and resolution. This enables tea functional component analysis technology to reach a new platform, realizing a rapid, high-throughput, high-sensitivity, and accurate quantitative analysis. It also solves the defects of traditional colorimetric spectrophotometry and chromatography that cannot be accurately quantified due to the interference of a complex matrix and similar component structures in tea.

This review aims at presenting the current state-of-the-art recent advances in analytical methods from the perspectives of determining total polyphenols and individual polyphenols in tea, illustrated with various examples. Additionally, the rapid analysis strategy for total polyphenols and the application of high-throughput HRMS in the targeted and non-targeted analysis of tea polyphenols are highlighted.

## 2. Types of Polyphenols in Tea

Polyphenol compounds, as the main secondary metabolites in plants, are important factors that determine the sensory and nutritional quality of tea, fruits, vegetables, and other plants [17]. The structures of these compounds include benzene rings with one or more hydroxyl groups, ranging from simple phenolic molecules to complex high-molecular polymers [18]. The diversity of this structure has led to a wide variety of polyphenols in natural ecosystems. More than 8000 polyphenols have been identified, and this number is still increasing [19]. There is no general classification of polyphenols, and some scholars have proposed their own classifications. For example, Manach et al. classified polyphenols in nature into four categories: phenolic acids, flavonoids, stilbenes, and lignans based on the number of phenolic rings and the structural elements of the connection between the rings [20]. However, according to the different content of polyphenols in tea, tea polyphenols are considered as a general term of (a) catechins, (b) flavonoids and flavonols, (c) anthocyanins, and (d) phenolic acids [21]. The structures of representative substances are shown in Figure 2. Except for phenolic acids, others have the basic skeleton of C6-C3-C6 configuration (a derivative of benzopyran or chromane), which essentially consist of two aromatic rings A and B, connected by a 3-carbon bridge, usually in the form of a heterocyclic ring C [22].

The most important part of tea polyphenols is the type of catechin, which accounts for about 70% to 80% of the total polyphenols in tea [21]. Catechins play an important role in the formation of the tea’s color, aroma, and taste quality. At the same time, the health function of tea mainly depends on these catechins, including, but not limited to, (−)-gallocatechin (GC), (−)-epigallocatechin (EGC), (+)-catechin (C), (−)-epicatechin (EC), (−)-epigallocatechin gallate (EGCG), (−)-gallocatechin gallate (GCG), (−)-epicatechin gallate (ECG), and (−)-catechin gallate (CG). Among them, EGCG is the most abundant substance, accounting for 50% to 75% of the total catechin content. Furthermore, EGCG has also been described as a second signal messenger, a stimulator of plasma membrane proteins, and a modulator of metabolic enzymes, playing a key role in the prevention of various diseases by tea [5]. The content of catechins varies greatly among different teas. The content of total catechins in green tea, yellow tea, and oolong tea is more than 80 mg/g, and in white tea is about 55 mg/g, but the content is below 20 mg/g in black tea and dark tea. The content of individual catechins is also the same, such as the content of EGCG in green tea that can reach more than 100 mg/g, while EC is less than 10 mg/g in the six major teas, and even less than 1 mg/g in black tea and dark tea [4]. In addition, some catechins are converted into other polyphenols during the unique manufacturing process of tea. For instance, the fermentation process of black tea can promote the oxidation of catechins, which are then condensed into theaflavins (TFs) and thearubigins (TRs), making black tea rich in TFs [23].

Flavonoids are polyphenol compounds containing 15 carbon atoms, and their typical structural characteristics are the existence of a carbonyl group at position four of the C ring in the C6-C3-C6 skeleton (Figure 2b). Flavonoids are the most abundant polyphenols in the plant kingdom. Some scholars even divide polyphenols into flavonoids and nonflavonoids [24,25]. The dietary flavonoids can be subdivided into flavonols, flavonoids, flavanones, and isoflavones. Among them, flavonols are the most common flavonoids in food. The representative flavonoids in tea are kaempferol, quercetin, myricetin, and so on. Moreover, flavonols in tea tend to combine with sugars to form flavone glycosides, such as rutin, quercetin-3-glycoside, etc. The content of flavonols and glycosides account for about 3–4% in tea (dry matter) [21].

Anthocyanin compounds are a kind of natural water-soluble pigment widely existing in plants, and their structures are highly conjugated and belong to chromogen derivatives, having anti-oxidant, anti-cancer, anti-inflammatory, and hypoglycemic functions [26]. There are six anthocyanins commonly found in nature: pelargonidin, cyanidin, peonidin, delphinidin, petunidin, and malvidin (Figure 2**c**). In addition, more than 500 anthocyanins have been isolated from plants. Generally, the anthocyanin content in tea (dry weight) accounts for about 0.01%, but in some special tea varieties (such as Zijuan), the anthocyanin content can be as high as 2.7–3.6% [27].

Phenolic acids account for about one-third of dietary polyphenols and exist in plants in free and binding forms [28]. Phenolic acids are a kind of aromatic compound with carboxyl and hydroxyl structures. They are composed of two subgroups of hydroxybenzoic acid and hydroxycinnamic acid. Typical representatives of hydroxybenzoic acid are gallic acid, *p*-hydroxybenzoic acid, protocatechuic acid, vanillic acid, and syringic acid, which have a common C6-C1 structure. Hydroxycinnamic acid is an aromatic compound with a C6-C3 structure, and caffeic acid, ferulic acid, *p*-coumaric acid, and erucic acid are the most common representatives [29]. The phenolic acid compounds contained in tea account for about 5% of fresh tea leaves (dry weight). Among them, gallic acid is the most representative phenolic acid in tea (1–2% of dry weight), its content is as high as 4.5 mg/g in fresh tea leaves [20,21].

There are many types of polyphenols in tea, and their contents vary greatly. In addition to the four major types of tea polyphenols classified above-mentioned in tea, there are many trace and unidentified tea polyphenols. Therefore, it is necessary to discuss the analytical methods of tea polyphenol compounds according to different analytical requirements. The determination of tea polyphenols could be divided into two categories according to different requirements: the determination of the total polyphenols and individual polyphenols in tea. As shown in Figure 3, the analytical methods for the total tea polyphenols include titration, spectrophotometry, near-infrared spectroscopy, electrochemical methods, etc. Among them, spectrophotometry is the most widely used, such as the application of international standard methods (ISO 14502-1:2005) and China’s national standard method (GB/T 8313-2018). Chromatography, however, is the most frequently used method for the qualitative and quantitative analysis of individual polyphenols in tea.

## 3. Quantification of Total Tea Polyphenols

Total tea polyphenols refer to all polyphenol compounds in tea. Polyphenols are ubiquitously spread through the plant kingdom, and there are many polyphenols that have not been discovered yet [30]. The detection of total tea polyphenols can avoid the ambiguity, which is rather available without the need for sophisticated equipment. Although the determination of total tea polyphenols inevitably has certain errors, it is widely used in tea research because of its feasibility in practical applications, especially for online rapid monitoring methods for total tea polyphenols [22]. Table 1 summarizes the applications on the analytical methods of total tea polyphenols in the past 10 years. Spectrophotometry is the most widely used analytical method for total tea polyphenols, but in recent years, more and more studies of total tea polyphenol determination focused on the non-destructive infrared spectroscopy and rapid electrochemical sensors analysis.

### 3.1. Titration

Among the analytical methods of total tea polyphenols, potassium permanganate oxidation titration is a classical method for the determination of the total tea polyphenols. Tea polyphenols are reductive substances and can be oxidized by potassium permanganate. With indigo red or indigo blue as indicators, the total amount of tea polyphenols is determined by the color change at the end point of titration. However, this method has obvious disadvantages: In addition to the polyphenols that can be oxidized, some non-polyphenols can also be oxidized (such as vitamin C); Secondly, it is difficult to master the end point of titration in experiments, resulting in a large probability of human error. Although this method has some defects in the determination of the total tea polyphenols, it is still applied in some tea industries due to the economical and convenient experimental conditions.

In the potassium permanganate oxidation titration method for the determination of total tea polyphenols, in addition to acidic indigo blue or indigo red as indicators, Yuan et al. developed a new potassium permanganate oxidation titration method for the determination of tea polyphenols by using 1.10-phenanthroline-iron (II) indicator [47]. The titration took potassium permanganate as the oxidizing agent in a strong sulfuric acid medium and the end point of titration was determined with a 1.10-phenanthroline-iron (II) indicator. At the end point of titration, the color changed from purplish red of potassium permanganate to colorless. A little too much titrant could make the solution immediately turn deep red, and the color change of the end point was sensitive. The recovery rate of this method was between 99.9% and 101.1%, and the RSD (relative standard deviation) was between 1.01% and 1.40% (*n* = 5).

As a traditional method, the titration method for the determination of the total tea polyphenols is not common. However, based on the principle of potassium permanganate oxidation, quantitative analysis combined with other analytical techniques is a feasible direction. Francis et al. developed an analytical method for the determination of oxidative capacity in teas and fruit juices by fluid injection analysis combined with chemiluminescence detection based on the oxidation reaction of potassium permanganate [48]. This study demonstrates the great potential of acid potassium permanganate chemiluminescence to explore antioxidants in complex sample matrices.

### 3.2. Spectrophotometry

The working principle of spectrophotometry is based on a certain linear relationship between the absorbance and concentration of the substance in a dilute solution. This method performs quantitative and qualitative analysis on substances by measuring the absorbance of the substance at a specific wavelength or the absorption spectrum in a specific wavelength range. Spectrophotometry is the most widely used analytical method to determine the total tea polyphenols. The key to this method is to select the most suitable chromogenic reagent and the appropriate measurement conditions. According to the difference of chromogenic reagents for the determination of tea polyphenols, there are two spectrophotometric methods: the Iron (II) D-tartrate colorimetric method and the Folin–Ciocalteu colorimetric method.

As early as 2003, China’s national standard method (GB/T 8313-87) for the determination of total tea polyphenols applied the Iron (II) D-tartrate colorimetric method. In this method, polyphenols can form a blue–purple complex with Iron (II) D-tartrate under a certain pH condition, and then be quantitated by a spectrophotometer. However, the application of this method to determine total tea polyphenols has certain defects. Different polyphenol components in tea have obvious differences in the color rendering ability of Iron (II) D-tartrate, which makes it difficult to acquire an accurate conversion coefficient for quantitative analysis. Moreover, there is an asymmetry of the absorption peak of the complex formed by tea polyphenols and Iron (II) D-tartrate, which also causes errors in the quantitative analysis.

The Folin–Ciocalteu colorimetric method is the most used method to determine the total polyphenols in plants. The existing international standard methods (ISO 14502-1:2005) and China’s national standard method (GB/T 8313-2018) for the determination of total tea polyphenols takes Folin–Ciocalteu reagent as the chromogenic reagent. In both methods, a 70% methanol aqueous solution is used to extract the tea polyphenols from ground tea samples in a water bath of 70 °C, and Folin–Ciocalteu reagent oxidizes -OH groups in tea extracts making the color of the extracts blue. The gallic acid is used as a calibration standard to quantify the tea polyphenols with the maximum absorption wavelength of 765 nm. These methods show good accuracy but have the disadvantage of complicated pre-treatments and the high detection costs. Currently, this method is still widely used in the determination of total polyphenols in tea and is often used as a standard to verify the accuracy of the developed method [49,50,51]. In addition to the two chromogenic reagents mentioned above, there are also new chromogenic reagents for the determination of total tea polyphenols by visible spectrophotometry. Al-Shwaiyat et al. used 18-MPC (18-molybdodiphosphate heteropoly complex) as a chromogenic reagent to determine the total tea polyphenols in green tea, showing a higher selectivity than Folin–Ciocalteu reagent [34].

### 3.3. Infrared Spectroscopy

The research of infrared spectroscopy began in the early 20th century. Since the advent of commercial infrared spectroscopy instruments in 1940, infrared spectroscopy has been widely used in analytical research [52]. Infrared spectroscopy determines the molecular structure and identifies compounds based on information of the relative vibrations between the atoms and molecular rotations. When a beam of infrared light with successive wavelengths passes through a substance and the vibration or rotation frequency of a group in the substance is the same as the frequency of the infrared light, the molecule absorbs energy to transition from the original ground state vibration (rotation) energy to a higher energy vibration (rotation) energy, resulting in the absorption of light at that wavelength by the substance. The absorption of infrared light by the molecule is recorded with an instrument to obtain an infrared spectrum. Infrared spectrograms take the wavelength (λ) or wave number (σ) as the abscissa to represent the position of the absorption peak, and the transmittance (T%) or absorbance (A) as the ordinate to represent the absorption intensity.

Nowadays, more and more studies have applied infrared spectroscopy technology to determine the total tea polyphenols because of the advantages of non-destructive detection, a low equipment cost, a fast spectral acquisition, and no chemical reagents [35,38,40]. Chen et al. initially explored the possibility of using near-infrared spectroscopy to quickly quantify the content of caffeine and total tea polyphenols in green tea, indicating that near-infrared spectroscopy combined with multivariate calibration analysis could be applied to the on-site and rapid monitoring of total tea polyphenols in the tea industry [53]. The prerequisite for the rapid quantitative analysis of total tea polyphenols by infrared spectroscopy is to establish a reliable calibration model. Xiong et al. established a quantitative method for the total tea polyphenols in Iron Buddha using near-infrared spectroscopy combined with a multi-spectral imaging system through the partial least squares (PLS) [40]. The results showed that the PLS method was considered as the best model for the determination of total tea polyphenols. Guo et al. compared different multivariate quantification methods combined with near-infrared spectroscopy analysis to effectively determine the total tea polyphenols in matcha, and investigated subsequent algorithms based on synergy interval partial least squares (SiPLS), including a successive projections algorithm (SPA), a genetic algorithm (GA), and simulated annealing (SA) [38]. The results showed that the correlation coefficients of siPLS-SA and SIPLS-SPA models were both higher than 97%, showing accurate and good prediction performances.

In addition to the advantages of non-destructive detection, fast spectral acquisition, and no need for chemical reagents for the determination of total tea polyphenols by near-infrared spectroscopy, the superiority of near-infrared spectroscopy is that it can realize the rapid detection of total tea polyphenols in different tea matrices (fresh tea leaves or raw materials in each step of processing), without the complicated and time-consuming fixing method (the rapid deactivation of enzymes in tea samples and the conversion to dry tea). Therefore, the determination of total tea polyphenols by near-infrared spectroscopy has a very broad application prospect in the tea industry.

### 3.4. Electrochemical Analysis

Electrochemical analysis is an important part of the instrumental analysis method. According to the electrochemical properties and behavior of substances in a solution, the relationship between electrochemical parameters (such as potential, conductivity, and current) and the concentration of target substances is used to achieve qualitative analysis [54]. Electrochemical analysis methods include potentiometric titration, stripping voltammetry, electrochemical sensors, polarography, etc. Among them, electrochemical sensors with a high accuracy and sensitivity, rapid analysis, miniaturization, and on-site detection advantages become a promising direction of electrochemical analysis, and have been widely used in the study of total tea polyphenols determination [55,56].

In electrochemical analysis, a suitable electrode modification material could better serve the electrochemical analysis technology. Based on the antioxidant properties of tea polyphenols (the ability to remove reactive oxygen species), Žegarac et al. used an unmodified glass carbon electrode (GCE) to determine tea polyphenols in acetate buffer (pH = 3) in commercially available flavored fruit teas. The results suggested that the ortho-dihydroxy-phenol and gallate group are the main contributors to the antioxidant capacity in fruit teas [57].

In addition to the traditional unmodified GCE, chemically modified electrodes (CMEs) get increasing interest because of their higher electron transfer rates, larger specific surface areas, and better catalytic performances in electrochemical analysis. Eguílaz et al. developed a sensor made of CME (modified by single-walled carbon nanotubes covalently functionalized with polytyrosine) for the high-sensitivity quantification of total tea polyphenols in tea extracts [33]. This method used gallic acid (GA) as the standard representing the total tea polyphenols. The results showed that the linear range of the GA current response calibration curve was 5.0 × 10^−7^ to 1.7 × 10^−4^ M, the detection sensitivity was 518 ± 5 m AM^−1^ cm^−^^2^, and the LOD was 8.8 nM. Besides, Shi et al. used CME modified by composite FeCF (cassava fibers and iron nanoparticles) as a biosensor to determine tea polyphenols [44]. The results showed that the biosensor could accurately quantify tea polyphenols in the concentration range of 3.5–31.5 µM, with a detection limit of 0.1 µM, showing excellent selectivity and stability, and could be used to detect tea polyphenols in actual beverage samples.

With the development of chemometrics, electrochemical analysis combined with chemometrics methods are expected to be applied to the rapid detection of tea polyphenols. Jiang et al. developed a homemade color-sensitive detector combined with multivariate analysis to determine total tea polyphenols in green tea infusions [39]. The sensor was prepared firstly to acquire the aroma information of green tea, and then the color components were extracted and optimized using an ant colony optimization (ACO) algorithm. Thus, the optimized color feature components were used to establish an extreme learning machine (ELM) model for the quantitative determination of tea polyphenols. The results showed that the correlation coefficient of the best ELM model was 0.8035, and the root mean square error prediction (RMSEP) was 1.6003% in the validation set.

Traditional methods for the determination of total tea polyphenols such as titration and spectrophotometry, are not only easily influenced by other reducing non-phenolic substances in the sample matrix, making the obtained result higher than the true value, but require a strict and complicated sample preparation. In the tea industry, it is of great significance to on-site monitor total tea polyphenols during tea planting, processing, brewing, and market circulation. Electrochemical sensors have the advantages of a high accuracy, a high sensitivity, miniaturization, and on-site quick testing, which are expected to be widely used in the analysis of total tea polyphenols.

### 3.5. Others

In addition to the above-mentioned methods commonly used in the determination of total tea polyphenols, some emerging rapid measurement techniques are gradually applied to analyze total tea polyphenols, such as paper-based sensor technology. Hidayat et al. developed a low-cost, portable, and disposable paper-based sensor for the determination of total tea polyphenols in green tea beverages [32]. The polyphenols sensor based on the mixture of NaIO_4_ and MBTH immobilized onto filter paper as a test strip had a response time of 9 min. The linear range of the developed sensor was in the range 25–300 ppm, and the detection limit of catechin was 10.2 ppm.

Based on paper-based sensors, incorporating automatic flow procedures can demonstrate greater advantages in terms of repeatability, automation, and operational robustness. More recently, Hao et al. developed a rapid determination method of total tea polyphenols by a kinetic matching approach on microfluidic paper-based analytical devices [46]. Based on the principle of the Folin–Ciocalteu reaction, the paper chip was filled with sodium carbonate solution among individual channels simultaneously, triggering all reactions at the same time (Figure 4). Gallic acid (GA) was found valid as a standard compound for the kinetic matching measurement of tea samples. This method took gallic acid as the standard compound for kinetic matching measurement and could determine the total tea polyphenols in a tea infusion within 10 min without any complexed time control procedure needed. The developed method showed a good linearity in the total tea polyphenols range of 10–100 mg/L (*r* > 0.9955) and the inter-chip precision was 5.6% (*n* = 11). Despite the accuracy of this method, it was poorer for some big leaf tea samples with colored matrices, but the method could be highly probable to on-site monitor total tea polyphenols during the tea production.

## 4. Simultaneous Determination of Individual Polyphenols

Different from the analytical methods of total tea polyphenols, the simultaneous determination of individual tea polyphenols puts forward higher requirements for sample pretreatment and instrumental separation and analysis. Due to the complexity of the tea matrix, the extraction, purification, and separation of individual tea polyphenols are difficult [58,59]. High-throughput and high-sensitivity qualitative and quantitative analyses of individual tea polyphenols usually requires relatively sophisticated and expensive analytical instruments. At present, liquid chromatography (LC) is the main method for the determination of individual polyphenols.

### 4.1. Liquid Chromatography (LC)

HPLC combined with an ultraviolet (UV) or a diode array detector (DVD) is the preferred method for the simultaneous separation and quantification of different polyphenols in tea. Since the introduction of HPLC in the 1970s, this technology has been widely used in the analysis of polyphenols, and hundreds of applications in food analysis have been published [22,25,30]. The technology of HPLC-UV was first employed to simultaneously determine the eight most abundant catechins in tea by Goto et al. in 1996 [60]. With the continuous optimization of analytical methods for polyphenols in tea, HPLC is still the mainstream method for the determination of individual tea polyphenols. According to the current international standard method (ISO 14502-2:2005), HPLC-UV is used to determine six polyphenols and caffeine simultaneously in green tea. The method was performed on reversed-phase C18 column (250 × 4.6 mm, 5 μm), with 9% acetonitrile (containing 2% acetic acid and 20 μg/mL EDTA) and 80% acetonitrile (containing 2% acetic acid and 20 μg/mL EDTA) as the water phase and organic phase, respectively, and detected by a UV detector at 278 nm. In this section, Table 2 summarizes the representative applications of simultaneous determinations of polyphenols in tea by HPLC over the past 10 years. The HPLC methods for the determination of tea polyphenols are mainly performed on reversed-phase C18 chromatographic columns, and the mobile phase generally employs a binary solvent system composed of acidified aqueous solutions and polar organic solvents (methanol or acetonitrile).

The obvious advantage of HPLC in the analysis of polyphenols lies in the simultaneous determination of multiple compounds, showing a satisfied separation performance and sensitivity. Zhao et al. developed a method for the determination of seven catechins and caffeine in Wuyi Rock tea using HPLC combined with a photodiode array detector (PDAD) [67]. This method used a reversed-phase C18 column (Zorbax Eclipse XDB-18, 4.6 × 250 mm, 5 μm) for separations, with the ratio of water/acetonitrile/acetic acid/EDTA of 888:90:20:2 (*v*/*v*/*v*/*v*) as the water phase and 178:800:20:2 (*v*/*v*/*v*/*v*) as the organic phase, and the sample pretreatment referred to ISO 14502-2 with appropriate adjustments. The application of this method is helpful to distinguish the two major rock tea varieties ‘Wuyi Rougui’ and ‘Wuyi Shuixian’.

The external standard method is the most frequently used quantitative method for the determination of polyphenols by HPLC. However, different tea plant varieties, different processed teas, and different matrices (Figure 1) present some challenges in chromatographic analysis, such as background drift (baseline drift or retention time drift), unknown interference, and high co-elution peaks, affecting the accuracy of qualitative and quantitative results. These interferences are almost impossible to completely eliminate by optimizing the stationary phase and elution program, so as to achieve sufficient resolution for all compounds. Therefore, it makes sense to correct the inevitable errors through mathematical methods. Zhang et al. compared three second-order calibration methods (including an alternate trilinear decomposition (ATLD) algorithm, multivariate curve resolution-alternating least squares (MCR-ALS), and alternating trilinear decomposition-assisted multivariate curve resolution (ATLD-MCR)) in processing LC data with retention time shifts and overlapping peaks [71]. The results showed that ATLD performed poorly with the average recovery being only 67.33–84.05% in the quantitative analysis of black tea and Cinacanthus nutans tea, while MCR-ALS and ATLD-MCR performed better with the average recoveries of 86.04–117.60% and 89.96–115.96%, respectively.

HPLC analysis of tea polyphenols is commonly performed on a reversed-phase C18 column with a length of 250 mm, an inner diameter of 4.6 mm, and is filled with octadecylsilane particles with inner diameters of 5 μm (Table 2). The determination of tea polyphenols in complex matrices by HPLC requires a high resolution and long time, which is a limitation for the high-throughput analysis of many samples. However, with the introduction of the concept of ultra-high-performance (pressure) liquid chromatography (UHPLC) by Jorgenson in 1997, UHPLC using sub-2 mm particle packed columns or porous-shell columns (with sub-3 mm superficially porous particles) opens a new possibility to improve the analytical method of complex matrices [76,77]. It can achieve separation speeds 5–10 times faster than traditional HPLC while improving resolution. The UHPLC analysis of polyphenolic compounds in a tea matrix is commonly used with the reverse phase C18 column with a length of 150 mm and an inner diameter of 2.1 mm, with the inner diameter of the packed particles being 1.7 μm. Compared with traditional HPLC, these short UHPLC columns achieved smaller particle stationary phases and higher linear velocities without sacrificing the separation effect. In addition, UHPLC has the advantages of higher resolution, higher peak efficiency, rapid separation, and reduced solvent consumption.

Guillarme et al. tested four different endcapped, deactivated reverse phase C18 columns packed with sub-2 µm particles (including a conventional C18 material (Hypersil Gold C18), a hybrid BEH C18 support (Acquity BEH C18), a hybrid BEH RP18 support with a polar (carbamate) embedded group (Acquity BEH Shield RP18), and a hybrid BEH phenyl material (Acquity BEH phenyl)) on the separation effect of eight main tea polyphenols [78]. The results are shown in Figure 5. The Acquity BEH Shield RP18 column (50 × 2.1 mm, 1.7 µm) (Figure 5D) could achieve the ideal separation effect of eight targets within 2 min, confirming that the endcapped, monomeric C18 columns are required for the separation of the naturally occurring tea catechins. Furthermore, it was demonstrated that columns containing a polar embedded group were beneficial to improve the selectivity of the structural isomer pairs of catechins.

### 4.2. Liquid Chromatography–Mass Spectrometry (LC–MS)

HPLC is suitable for the determination of the most abundant tea polyphenols such as GC, EGC, C, EC, EGCG, GCG, ECG, CG, etc. (unit with mg/g) [65,66,67]. However, it is difficult for UV or DAD detectors to achieve high sensitivity requirements for the trace polyphenols in tea. Mass spectrometry (MS) is especially suitable for the trace analysis of polyphenols with a low concentration in complex matrices because of its great advantages in sensitivity, precision, and resolution [25]. LC–MS combines the advantages of separation ability of LC with the strong identification capability of MS, which is an effective technology to analyze complex organic mixtures. It can provide a large amount of information about complex mixtures, allowing qualitative and quantitative of hundreds of components through a single analysis [79]. There are many studies involving the analysis of polyphenols using LC–MS, and some reviews and book chapters dedicated to this topic can be found in literatures [25,30]. Compared with single-stage MS, MS/MS provides higher selectivity and sensitivity, and can significantly simplify complex and laborious sample preparations, and has therefore become the main analytical technology [80]. The representative applications of tea polyphenols determination using LC–MS in the past 10 years are summarized in Table 3. UHPLC combined with MS (UHPLC-MS/MS) is the most widely used technique of tea polyphenols analysis, showing a good separation effect, realizing a high-throughput, with high sensitivity analysis of tea polyphenols.

Wu et al. compared two methods of HPLC-PDA and UHPLC-MS/MS to determine the performance of six flavonoids in fresh tea leaves, wherein HPLC-PDA applied a 4 μm Phenomenex Synergi Fusion-RP 80A column and UHPLC-MS/MS used a 2.6 μm Phenomenex Kinetex XB-C18 100A (Torrance, CA, USA) column [87]. The results showed that the standard curves of six flavonoids have good linear correlation (*R*^2^ > 0.99) in a certain concentration range by two analytical methods, and the LODs and LOQs obtained by UHPLC-MS/MS were less than 0.01 and 0.03 ng, respectively, three quantities lower than the quantitative method of HPLC (LODs < 2.43 ng and LOQs < 7.68 ng).

Mass spectrometry is an analytical method for measuring mass-to-charge ratio (*m*/*z*). An electrospray ionization source (ESI) is the most commonly used ionization technology. Tea polyphenols in the ESI source should typically be ionized in a negative ion mode to produce deprotonated molecular ions (M-H)^−^ (Table 3). It is worth noting that tea is a highly complex matrix, enriched with high amounts of alkaloids, polyphenols, and some of the unique tea pigments, such as theaflavins and thearubigins, resulting in polyphenols targets suffering from the interferences of coeluting compounds, leading to the ionization suppression or enhancement in the electrospray ionization (ESI) source [88,89]. This effect is obvious in matrices of dry tea and fresh leaves. Thus, purification is often required to extract and purify the targets during sample preparation in the analysis of tea by LC–MS. Currently, different sample preparation methods such as solid-phase extraction (SPE) or QuEChERS combined with LC–MS have been widely used in tea matrix purification [90,91]. However, paradoxically, adsorbent materials commonly used for purification in tea, such as dispersed solid-phase adsorbent PVPP, primary secondary amines (PSA), C_18_, GCB, Al_2_O_3_, and Florisil, all show adsorption effects on tea polyphenol targets we discussed [92,93]. Especially for PSA, as a weak anion exchange agent, there is a good adsorption of polar compounds such as fatty acids, organic acids, anthocyanins, and plant polyphenols [94]. Therefore, these common purification treatments are not common in LC–MS-based tea polyphenols analysis methods, and another solution is diluting the extracts to weaken the matrix effect. Long et al. developed a UHPLC-ESI-MS method for the determination of six flavonol glycosides and seven procyanidins in pu-erh tea pile-fermentation with a biomass to solvent ratio of 1:200, to reveal the substances changes during the key step of pu-erh tea processing [11]. The separation was performed on a Waters Acquity Beh Sheid C18 column (2.1 × 50 mm, 1.7 μm), with 0.1% formic acid water as the aqueous phase and acetonitrile as the organic phase. The targets were extracted with 70% (*v*/*v*) methanol–water by ultrasonic extraction. Liu et al. also developed a UHPLC-ESI-MS/MS method for the determination of four primary catechins, including EC, EGC, ECG, and EGCG in fresh tea leaves [81]. The pretreatment of this method took the fixing method of a pulsed electric field to transform fresh tea leaves into dry tea, and the targets were extracted using 50% acetone/water (*w*/*w*) with a biomass to solvent ratio of 1:100. The separation was performed on an Acquity UHPLC BEH C18 Column (150 × 2.1 mm, 1.7 μm), with 0.1% (*v*/*v*) formic acid water as the aqueous phase and 0.1% (*v*/*v*) formic acid acetonitrile as the organic phase.

In addition to the two matrix forms of fresh tea leaves or dry tea, there are also tea samples in the form of tea infusions (Figure 1). The matrix effect of tea infusions is relatively weak, because it is actually a disguised form of tea sample with an exaggerated dilution ratio. Olech et al. developed a method for the determination of 10 phenolic acids and 10 flavonoid glycosides in rose tea (tea infusion) using UHPLC-ESI-MS/MS [84]. The method applied an Acquity HSS T3 column (1.0 × 100 mm; 1.8 µm) with 0.1% formic acid water as the aqueous phase and acetonitrile as the organic phase, which achieved the high sensitivity analysis of 10 phenolic acids and 10 flavonoid glycosides in the rose infusion. The standard curves for different concentration levels range from 0.05–15 ng/μL.

Triple quadrupole (QqQ) mass analyzers are recommended for the LC–MS quantitative analysis of polyphenols in tea, exhibiting a high sensitivity in MRM mode. The QqQ consists of two scanning quadrupole modules Q1 and Q3, which are located on both sides of the collision cell Q2 used for collision-induced dissociation (CID). The system has four monitoring modes, including parent ion scan, product ion scan, neutral loss scan, and the most sensitive selective reaction monitoring (SRM). The last one is used for quantification, and the others are used to identify the analytes. In the SRM mode, by optimizing the collision voltage (CE) of two or more selective ion pairs for each compound, the two or more selective ion pairs are qualitatively analyzed, among which the most sensitive selective ion pair is used for quantitative analysis.

Bao et al. have done a lot of work in the separation and identification of new tea polyphenols, and LC-QqQ-MS was often used for the qualitative and quantitative analysis of these new polyphenols [12,16,85,95,96]. For example, they synthesized four hydroxycinnamoylated catechins (HCCs) standards in tea [16]. The total ion chromatogram (TIC), extracted ion chromatograms and MS/MS spectra of the targets in UHPLC-QqQ-MS/MS are shown in Figure 6. Four HCCs in 20 tea varieties were quantitatively analyzed using UHPLC-QqQ-MS/MS. It was found that HCCs are widely present in two major tea varieties *Camellia sinensis* var. *sinensis* (CSS) and var. *assamica* (CSA), the total content of the four HCCs and EGC-pC in CSS is significantly higher (the total amount of HCCs is 729–1398 mg/kg, EGC-pC is 483–1052 mg/kg), while the EC-C in CSA is significantly higher (65–176 mg/kg), indicating that EC-C and EGC-pC can be used as biochemical markers to distinguish the two major types of tea varieties CSS and CSA.

### 4.3. HRMS

Polyphenol compounds typically contain a C6-C3-C6 skeleton with a variety of sub-groups on a specific carbon position of the molecule. Structural diversity originates from the biosynthesis and distribution of different plant species [22,25]. Then, tea manufacturing manners, such as thermal treatments, fermentation, brewing, and storage, etc., can also promote the interaction of many components and change the existing forms of polyphenols [4,23,97]. The structural diversity and changes of the polyphenol compounds hinder its accurate qualitative and quantitative analysis.

The resolution of the mass spectrometer plays an important role in the analysis of the compounds in tea samples, and the high-resolution mass analyzer can obtain the information of the elemental composition of the analytes. Different mass analyzers in LC–MS such as QqQ, Q-TOF, and Q-Orbitrap have been used to analyze polyphenols in tea (Table 3). Tandem mass spectrometry (QqQ) is a mature technology, but the selectivity provided by its unit resolution has become a liability. In recent years, with the development of HRMS and its impact on traditional MS/MS, increasing the number of HRMS have replaced traditional MS, such as magnetic mass spectrometry, TOF-MS, orbitrap mass spectrometry (Orbitrap-MS), and fourier-transform ion cyclotron resonance mass spectrometry (FTICR-MS), etc., [80,98]. These HRMS have a resolution of more than 10,000, and the mass tolerance is usually less than 5 ppm. HRMS make the analysis of analytes more accurate by multi-stage scanning combined with library searching, which can effectively distinguish mixtures and greatly reduce the requirement of preparation. Compared with traditional MS/MS, HRMS also reduces the requirement of chromatographic separation and can analyze hundreds of compounds at the same time, which is an extremely effective method for the trace analysis of multiple polyphenols in complex matrices. HRMS is characterized not only by its high sensitivity, excellent resolution, and wide dynamic range of detection, but also by its ability to be combined with chromatographic techniques for online structural identification of individual peaks. High mass resolutions enable the discrimination of metabolites from nominally isobaric biotransformations or background interference [99]. Traditional magnetic mass spectrometry or FTICR-MS is too slow to perform, complicated to operate, and may be expensive to purchase and maintain [80]. At present, TOF and Orbitrap are the two most common high-resolution mass analyzers that combine LC to analyze polyphenols in food [100]. Based on this, this review only covers two modern TOF-MS and Orbitrap-MS that can satisfy the high throughput, high sensitivity, high mass resolution, and sufficient dynamic range of tea polyphenols.

#### 4.3.1. TOF-MS

UHPLC-TOF is an earlier commercialized high-resolution mass spectrometer. TOF-MS is currently the most reported HRMS, and the number of related studies significantly exceeds that of Orbitrap [80]. In particular, the hybrid Quadrupole TOF (Q-TOF) system has been proven to be of great value in the qualitative and quantitative analysis of targeted and untargeted metabolites in complex matrices due to its fast acquisition speed, full scan sensitivity, high resolution, and accurate quality detection capability.

Among the various newly discovered tea polyphenols, one of the most studied polyphenol subgroups is flavonoids, which may come from the interaction between the main catechins and theanine (the main amino acid in tea). However, these compounds have multiple stereogenic centers to form multiple isomers, which makes accurate analysis of these isomers difficult. Zhang et al. developed a method to qualitatively and quantitatively analyze seven flavoalkaloids in different tea plant varieties and different processing stages using UHPLC-Q-TOF-MS [12]. The chromatographic separation was performed on an ACQUITY UPLC HSS T3 column (2.1 × 100 mm, 1.8 µm), with 0.1% formic acid in water as the aqueous phase and acetonitrile as the organic phase. In the positive ion mode, these seven targets consist of two molecular weight isomer compounds (three [M+H]^+^ at *m/z* of 570.1612 and four [M+H]^+^ at *m/z* of 554.1662 isomers) that were successfully separated. The results showed that the two tea varieties Yiwu and Bulangshan had the highest flavoalkaloid contents (total flavoalkaloid contents were 3063 µg/g and 2727 µg/g, respectively), and the contents of six flavoalkaloids in the nine tea varieties were all higher than 198 µg/g.

In addition, Zhang et al. also used UHPLC-Q-TOF-MS combined with molecular networks to discover and separate phenylpropane-substituted ester-catechins for the implication of the reaction mechanism among polyphenols during green tea processing [14]. The study initially identified 14 ester-type catechins (PSECs) through the creation of hypothetical PSEC data sets, MS/MS data acquisition, and molecular network construction. Most of the precursor ions of the Xi-Gui extract fraction (53 data) are distributed in clusters. Compared with the hypothetical PSEC data set, the maximum cluster marked in the red circle contains 23 precursor ions with a molecular weight deviation of less than 20 ppm. The 14 Da and 16 Da mass difference between the parent ions indicates that -CH_3_ and -OH substitutions have occurred on the molecule, which highly indicates the existence of part of the hypothetical PSEC molecular structure. This research provides an example for exploring new functional components of food sources based on a HR-MS/MS data network and provides new insights into the reaction mechanism of new catechin conjugates formed between polyphenols in green tea.

#### 4.3.2. Orbitrap-MS

Although TOF-MS is significantly higher than Orbitrap-MS in the number of published HRMS literatures, the growth rate of Orbitrap-MS studies is faster than TOF-MS [80,98]. Since Thermo Electron Corporation (Massachusetts, USA) launched the first Orbitrap-based commercial mass spectrometer (LTQ Orbitrap™) in 2005 [101], Orbitrap-based high-resolution mass spectrometry has become a very important analytical technology, and the mass resolution of 50,000 is easily achievable for all Orbitrap-based mass spectrometers. A key and challenging step in the implementation of the Orbitrap analyzer is the development of an external storage device, later called C-trap, that allowed ions to be accumulated before being injected into the Orbitrap analyzer, thereby allowing discontinuous Orbitrap analyzers to be connected to continuously operating ion sources (such as ESI) [102]. The successful combination of the Orbitrap analyzer and ESI source enables the Orbitrap mass analyzer to analyze complex mixtures. However, it is still necessary to control the size of the ion group during the analysis to avoid the serious space charge effect that occurs in the C-trap.

There are currently three series based on Orbitrap-MS, namely LTQ Orbitrap (Ion trap/Orbitrap hybrid mass spectrometer), Q Exactive (Quadrupole/Orbitrap hybrid mass spectrometer), and Orbitrap Fusion (Quadrupole/Orbitrap/ion trap tribrid mass spectrometer) [102]. Among them, liquid chromatography Q-Orbitrap MS has been popular in the past 10 years with the introduction of the Q-Exactive series, which combines the high-quality resolution provided by the Orbitrap mass analyzer with the high selectivity of the quadrupole, making it ideal for the high-throughput screening analysis of metabolomics and secondary metabolites [98]. Compared with Q-TOF MS, Orbitrap-based MS has a higher sensitivity and a more accurate mass in full scan mode, which is very important for the screening and identification of unknown compounds [103]. Therefore, Orbitrap MS can be regarded as a powerful technology for the targeted and non-targeted analysis of polyphenols in complex matrices.

Nardin et al. used an online SPE UHPLC system combined with Q-Exactive orbitrap MS to perform the targeted and non-targeted analysis of alkaloids in herbs [104]. In the positive ion mode, the mass spectrometer could accurately quantify 35 alkaloids and identify another 305 alkaloids in the herb extract (LODs were 0.04–10 μg/L) in the full MS-dd MS/MS scan mode with a resolution of 140,000, which highlights the capability of the Q-orbitrap MS in the analysis of plant natural products.

In tea research, Q-Orbitrap MS has gradually become a powerful tool for tea metabolomics analysis. Xu et al. revealed the influence of time scale and storage environment on the metabolites and taste quality of pu-erh tea based on the metabolomics of UHPLC Q-Exactive Orbitrap MS and the global natural product social molecular network [15]. The resolution was set to 70,000 in full MS mode with a scanning range of 100–1500 (*m*/*z*), and the resolution was set to 17,500 in dd-MS^2^ mode. The MS/MS spectra of 7471 features were obtained by UHPLC Q-Exactive Orbitrap MS and analyzed based on multivariate statistics, global natural product social (GNPS) multilibraries matching, and SIRIUS calculations. It was found that N-ethyl-2-pyrrolidinone-substituted Flavan-3-ols (flavoalkaloids), flavan-3-ols, and flavonol-*O*-glycosides are three characteristic polyphenols.

Xin et al. distinguished the changes of chemical components in Yunnan and Hunan green tea using UHPLC Q-Exactive Orbitrap MS [105]. The resolution was set to 35,000 in the full MS mode with a scanning range of 100–1500 (*m*/*z*), and the resolution was set to 17,500 in the dd-MS^2^ mode. Finally, 75 common tea chemical components including alkaloids, amino acids, catechins, flavonoids, flavonoid glycosides, phenolic acids, and theaflavins in green tea from two regions were identified by UHPLC Q-Exactive Orbitrap MS, of which, 29 components were identified by commercially available standard products, and the other 46 compounds were confirmed by a Q-Exactive Orbitrap MS non-targeted analysis strategy.

## 5. Conclusions and Future Trends

Tea polyphenols are the main functional components in tea, which have positive effects on the human health. At the same time, tea polyphenols also determine the unique quality and flavor of tea. Accurate analytical technology is of great significance for the in-depth analysis of tea polyphenols, the promotion of tea standardization, and the development of the tea industry.

For the determination of total tea polyphenols, spectrophotometry is the current mainstream analysis method. Folin–Ciocalteu chromogenic reagent, which replaces Iron (II) D-tartrate, has a better stability and meets the current quantitative analysis of total tea polyphenols to a certain extent. However, this method is based on oxidizing the -OH groups in tea polyphenols, which can easily oxidize some non-tea polyphenol substances, making the theoretical value of the result higher than the real value. Although the process of spectrophotometric determination of total tea polyphenols is relatively complicated, the rapid detection of total tea polyphenols has broader application prospects. Near Infrared spectroscopy technology with non-destructive detection and electrochemical sensor technology capable of rapid on-site detection have been increasingly used in the research of total tea polyphenol determination. Especially for the fast electrochemical sensor technology, which has a very broad application prospect in the aspects of rapid analysis, quality control, and the portable and remote control of tea samples in actual tea production, due to its advantages of high accuracy, high sensitivity, miniaturization, and on-site quick testing.

HPLC-UV(/DAD) is a suitable analytical method for the high content of individual tea polyphenols, especially catechins. Although mass spectrometry shows a high sensitivity compared with UV/DAD detectors, the high content of tea polyphenols easily causes serious matrix effects on ion sources in mass spectrometry, such as ESI. For low-concentration levels of tea polyphenols, UHPLC-MS/MS is obviously a useful tool for qualitative and quantitative analysis of targets, meeting the requirements of sensitivity and selectivity. However, it is still necessary to avoid the problem of matrix effects in the analysis process. It is a common way to sacrifice sensitivity by expanding the sample dilution factor. Therefore, the development of specific purification materials such as metal organic framework compounds (MOFs) and functional graphene materials is of great significance for the UHPLC-MS/MS analysis of tea polyphenols.

HRMS technologies, such as Q-TOF and Q-Orbitrap, are increasingly used in the analysis of tea metabolomics because of their fast acquisition speed, high full-scan sensitivity, and high mass resolution. HRMS has been proven to be of great value in the discovery of new tea polyphenols, isomer discrimination, marker screening, and the qualitative analysis of targeted and non-targeted metabolites in complex matrices. On this basis, derivatization and isotopic labeling will be more conducive to the identification of tea polyphenols.

## Figures and Tables

**Figure 1 foods-11-01425-f001:**
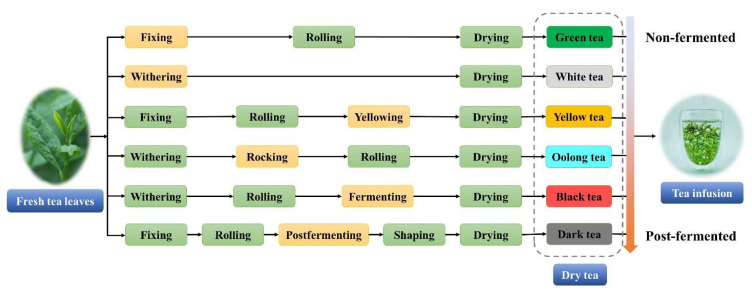
Manufacturing processes of six major tea categories: from fresh tea leaves to infusion.

**Figure 2 foods-11-01425-f002:**
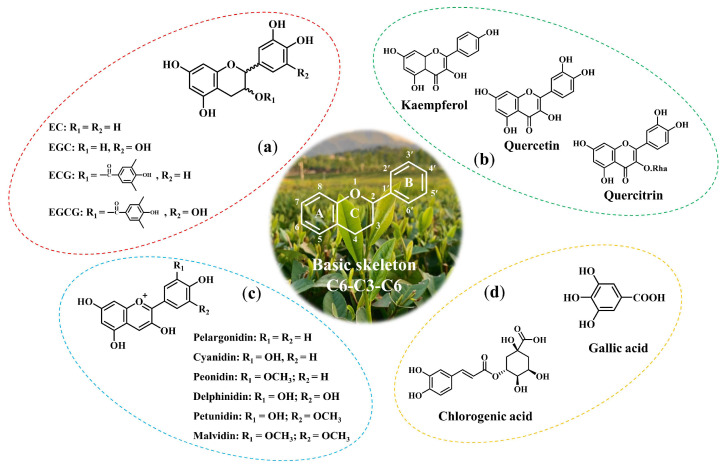
Structures of main polyphenols in tea. (**a**) catechins, (**b**) flavonoids and flavonols, (**c**) anthocyanins, and (**d**) phenolic acids.

**Figure 3 foods-11-01425-f003:**
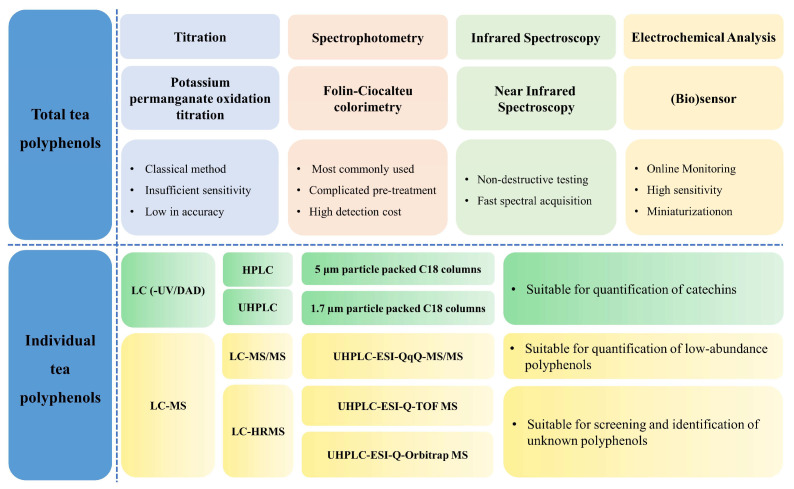
Summary of analytical methods for tea polyphenols (including representative applications and evaluations).

**Figure 4 foods-11-01425-f004:**
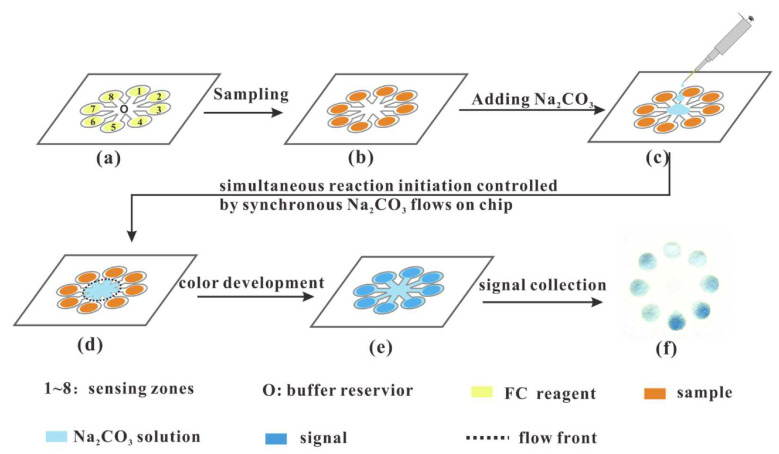
Schematic diagram for the process of rapid determination of total tea polyphenols with paralleled reaction and simultaneous signal acquisition. (**a**): Chips containing FC reagent, (**b**): Add samples, (**c**): Add Na_2_CO_3_ in the central region of the chip, (**d**): Simultaneously trigger the FC reaction, (**e**): Obtain the reaction signal, (**f**): Collect signals. Reprinted with permission from ref. [46]. Copyright 2021 Elsevier.

**Figure 5 foods-11-01425-f005:**
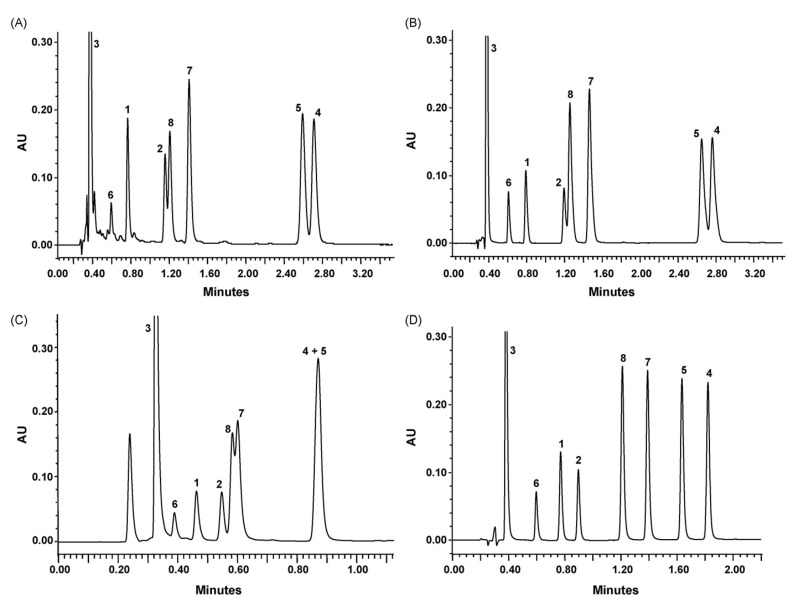
Optimal UHPLC chromatograms of an eight standard polyphenols mixture at 20 g/mL. (**A**) Acquity BEH C18 50 × 2.1 mm ID, 1.7 µm column. (**B**) Hypersil GOLD C18 50 × 2.1 mm ID, 1.9 µm column. (**C**) Acquity BEH phenyl 50 × 2.1 mm ID, 1.7 µm column. (**D**) Acquity BEH Shield RP18 50 × 2.1 mm ID, 1.7 µm column. 1: C, 2: EC, 3: GA, 4: CG, 5: ECG, 6: EGC, 7: GCG, 8: EGCG. Reprinted with permission from ref. [78]. Copyright 2010 Elsevier.

**Figure 6 foods-11-01425-f006:**
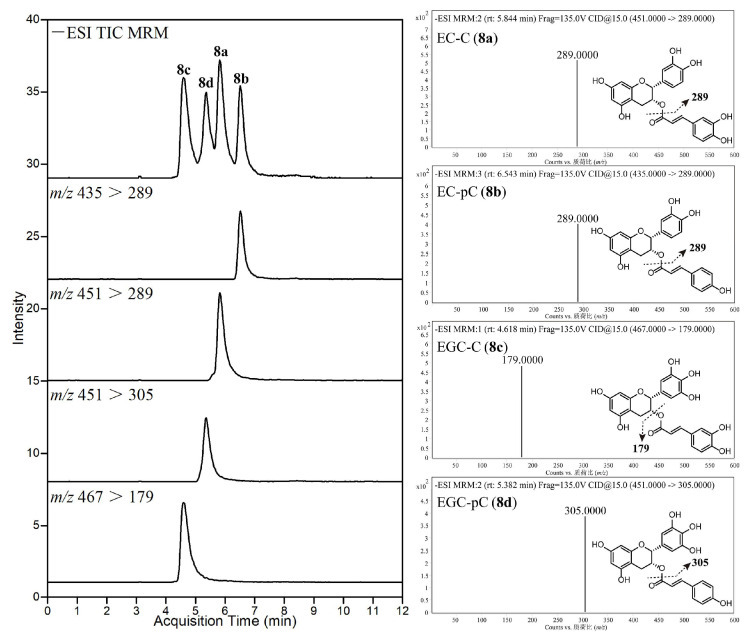
UHPLC-QqQ-MS/MS analysis of the four hydroxycinnamoylated catechins (HCCs) standards. Reprinted with permission from ref. [16]. Copyright 2021 Elsevier.

**Table 1 foods-11-01425-t001:** Recent applications of determination of total polyphenols in tea.

Tea Types	Analytical Method	Extraction	LODs	References
Dark tea	Hyperspectral imaging technology	Extraction with 70% methanol solution by water bath heating.	-	[31]
Green Tea	Paper based sensor	-	10.2 ppm	[32]
Green, black, classic, and herbal teas	Electrochemical (bio)sensors	Extracted using boiling water and then centrifuged to get the supernatant.	8.8 nM	[33]
Green Tea	Spectrophotometry	Extracted using boiling water.	5 × 10^−7^ mol/L	[34]
Fresh tea leaves	Infrared spectroscopy	0.01 g sample powder was mixed with 0.49 g KBr and extruded by a tablet machine.	-	[35]
Black tea	Voltammetric biosensor	3.0 g of tea samples were extracted with 50 mL of acetate buffer solution.	0.11 mM	[36]
-	Constant-current coulometry	2 g of tea samples were extracted with 200 mL boiling water for 3 min.	-	[37]
Green tea	Near-infrared spectroscopy	0.2 g of tea samples were extracted with 5 mL of 70% (*v*/*v*) methanol solution at 70 °C for 25 min.	-	[38]
Green tea	Homemade color-sensitive sensor	3 g of tea samples were extracted with 150 mL boiling water for 5 min.	-	[39]
Oolong tea	Near-infrared spectroscopy and multispectral imaging system	Without pre-treatment.	-	[40]
Green tea	Sensor	1.5 g of tea samples were extracted with 100 mL water heated up to 90 °C for 15 min.	0.6 µmol/L	[41]
-	A flow injection-fiber optic spectrophotometric system	1 g of tea samples were extracted with 40 mL of 60% ethanol by ultrasonic for 30 min.	0.01–0.10 mg/mL	[42]
Green tea	polyphenol sensor	1.5 g of tea samples were extracted with 100 mL of water heated at 90 ˚C for 15 min.	1.76 × 10^−7^ mol/L	[43]
-	Electroanalytical sensor	Without extraction.	0.1 µM	[44]
Green tea, Black tea, Dark tea	CdTe quantum dot fluorescence	Without extraction.	0.63 nM	[45]
Green tea, and black tea	Microfluidic paper-based analytical devices	Without extraction.	10.0–100.0 mg/L	[46]

Note: "-" refers to the relevant content not reported in the reference.

**Table 2 foods-11-01425-t002:** Recent applications of HPLC analysis of individual polyphenols in tea.

Targets	Chromatographic Separation	Mobile Phase	Sample Extraction	LODs	References
5 polyphenols and caffeine	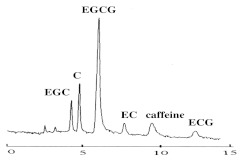	30% methanol containing 0.1% acetic acid	1 g of tea samples were boiled using 180 mL of water for 1 h	1.8–24 mg/L	[61]
10 polyphenols and caffeine	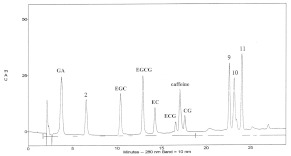	A (3% acetic acid); B (methanol)	1.9–3.8 g of tea samples were extracted three times with 20 mL 80% methanol for 3 h and then two times with 20 mL 80% methanol containing 0.15% HCl for 3 h.	>0.19 mg/L	[62]
5 polyphenols and caffeine	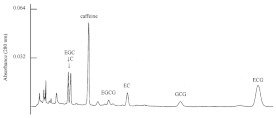	A (50 mM sodium phosphate (pH 3.3) and 10% methanol); B (70% methanol)	1 g of tea samples were extracted with boiling water (100 mL) for 30 min or 75% ethanol (100 mL) at 60 °C for 30 min.	-	[63]
13 polyphenols and caffeine	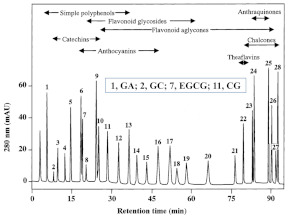	methanol/distilled water/formic acid (19.5:82.5:0.3, *v*/*v*/*v*)	50 mg tea powders were extracted with 2 mL of 90% methanol containing 0.5% acetic acid.	≥1 µM	[64]
24 tea constituents	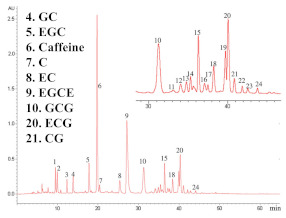	A (90:20:2:888, *v*/*v*/*v*/*v*; acetonitrile, acetic acid, EDTA, water); B (800:20:2:178, *v*/*v*/*v*/*v*; acetonitrile, acetic acid, EDTA, water)	2 g of tea samples were extrated with 100 mL of water.	-	[65]
15 phenolic antioxidants	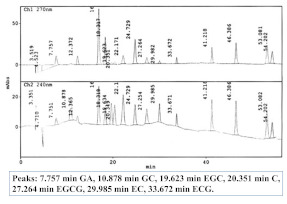	A (methanol–ACN (95:5, *v*/*v*); B (0.01% acetic acid)	2 g of tea samples were extracted with 100 mL water for 5 min in an ultrasonic bath.	0.8–3.0 mg/L	[66]
7 polyphenols and caffeine	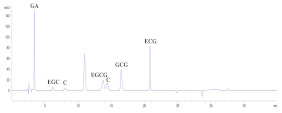	A (888:90:20:2, *v*/*v*/*v*/*v*; water, acetonitrile, acetic acid, EDTA); B (178:800:20:2, *v*/*v*/*v*/*v*; water, acetonitrile, acetic acid, EDTA)	0.2 g of tea leaves were extracted with 5 mL of 70% (*v*/*v*) aqueous methanol at 70 °C by a vortex mixer.	-	[67]
13 polyphenols, theobromine, and caffeine	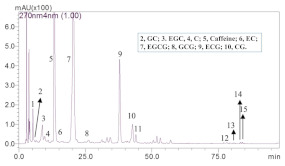	A (1% *w*/*v* orthophosphoric acid); B (acetonitrile)	0.2 g of tea powders were extracted overnight with 80% (*v*/*v*) aqueous methanol by an automatic mixer.	≥ 0.01 mg/mL	[68]
9 polyphenols and caffeine	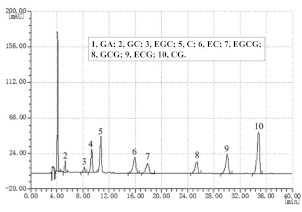	A (900:1:100, *v*/*v*; water, phosphoric acid, acetonitrile), B (200:1:800, *v*/*v*; water, phosphoric acid, acetonitrile)	10.0 g of powders were extracted with 50 mL boiled water and heated (100 °C) for 50 min.	-	[69]
17 polyphenols and three alkaloid compounds	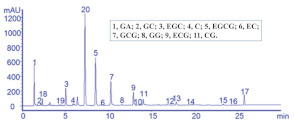	A (5% ACN, 0.261% ortho-phosphoric acid); B (80% MeOH)	1 g of powders were extracted with 44 mL MeOH hydrochloric acid (40:4, *v*/*v*) in an 85 °C water bath for 90 min by a reflux condenser.	0.0261–0.059 ng	[70]
5 polyphenols	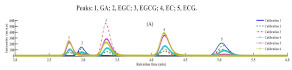	A (0.1% formic acid); B (MeOH)	0.2 g of tea powders were extracted with 10 mL boiling water in a water bath at 90 °C in dark for 10 min.	-	[71]
14 phenolic acids	-	A (methanol); B (1% acetic acid; *v*/*v*)	Without extraction	0.05–0.68 µg/L	[72]
9 polyphenols	-	A (0.1% formic acid in water); B (acetonitrile)	1 g of matcha powders were extracted with 100 g water (75 °C) stirring at 750 rpm for 1 min.	-	[73]
10 polyphenols and caffeine	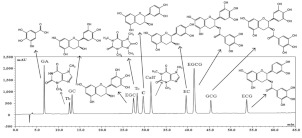	A (acetonitrile); B (water with 0.1% formic acid)	0.1 g of tea powders were extracted with 15 mL 90 °C water in a hot water bath for 40 min.	14.5–47.0 ng/mL	[74]
10 polyphenols and caffeine	-	A (methanol); B (0.2% formic acid)	0.1 g of tea samples were extracted with 4 mL of 70% (*v*/*v*) methanol-water under ultrasonic extraction for 10 min two times.	-	[11]
10 polyphenols and caffeine	-	A (0.17% aqueous acetic acid); B (acetonitrileA)	0.1 g of tea samples were extracted thrice with 3 mL of 70% (*v*/*v*) methanol–water by ultrasonic extraction for 10 min.	-	[75]

Note: "-" refers to the relevant content not reported in the reference.

**Table 3 foods-11-01425-t003:** Recent applications of LC–MS analysis of individual polyphenols in tea.

Tea Types	Targets	Analytical Method	Chromatography Column	Mobile Phase	Solid-Liquid Ratio (g/mL)	References
Fresh tea leaves	Four primary catechins, including EC, EGC, ECG, and EGCG	UHPLC-ESI-MS/MS	Acquity UHPLC BEH C18 column (150 × 2.1 mm, 1.7 μm)	A (0.1% formic acid); B (acetonitrile containing 0.1% formic acid)	1:100	[81]
Black tea	Gallic acid, p-Coumaric acid, Vanillic acid, Caffeic acid, Ferulic acid, Syringic acid, Sinapic acid	UHPLC-DAD-QA-ESI-TQTOF-MS	Phenomenex Gemini-NX column (150 × 2 mm; 3 μm)	A (0.1% formic acid in water); B (acetonitrile)	1:10	[82]
Green tea	Selenoaminoacid and catechin(MeSeCys, SeMet, SeCys, Catechin, Epicatechin, EGCG)	HILIC-MS/MS	Waters Atlantis HILIC (100 × 2.1 mm, 3 µm)	A (methanol); B (8 mM ammonium acetate pH 7 (85/15, *v*/*v*))	1:50	[83]
Dark tea	6 flavonol glycosides and 7 procyanidins	LC-QQQ-MS	Waters ACQUITY BEH Sheid RP18 column (2.1 × 50 mm, 1.7 μm)	A (0.1% formic acid-water); B (acetonitrile)	1:100	[11]
	eight puerins	LC-Q-TOF-MS	Waters ACQUITY BEH Sheid RP18 column (2.1 × 50 mm, 1.7 μm)			
Flower tea	10 phenolic acids and 10 flavonoid glycosides	UHPLC-ESI-MS/MS	Acquity HSS T3 column (1.0 × 100 mm; 1.8 µm)	A (0.1% aqueous formic acid); B (acetonitrile)	1:50	[84]
White tea	four flavoalkaloids: two novel pure compounds and a mixture of two isomers	UHPLC-ESI-MS/MS	ACQUITY UPLC BEH Shield RP18 column (2.1 × 150 mm, 1.7 μm)	A (0.1% aqueous formic acid); B (0.1% formic acid acetonitrile)	1:40	[85]
Green tea	14 ester type of catechins (PSECs)	UPLC-Q/TOF-MS	ACQUITY UPLC HSS T3 column (2.1 × 100 mm, 1.8 μm)	A (water containing 0.1% formic acid); B (ACN containing 0.1% formic acid)	1:40	[14]
Black tea	25 compounds were identified as apigenin glycosides, quercetin glycosides, kaempferol glycosides, theaflavins, theasinensin, and galloylglucoses	HPLC–MS/MS	Luna 5 m Phenyl-Hexyl C18 column (250 × 4.6 mm, 5 m)	A (0.1% formic acid in water with 5% methanol); B (0.1% formic acid in methanol with 5% water)	1:60	[86]
Fresh tea leaves	60 phenolic compounds were tentatively identified	UHPLC-Q-Orbitrap-MS	Acquity UHPLC Ethylene Bridged Hybrid (BEH) C18 column (150 × 2.1 mm, 1.7 μm)	A (0.1% formic acid in water); B (0.1% formic acid in acetonitrile)	1:40	[13]
Green tea	Five new flavoalkaloids and one new naturally occurring natural product were detected and isolated	UPLC-ESI-MS/MS	ACQUITY UPLC BEH Shield RP18 column (2.1 × 150 mm, 1.7 μm)	A (0.1% aqueous formic acid); B (0.1% formic acid acetonitrile)	1:40	[83]
Fresh tea leaves	15 flavonol glycosides	UPLC-QQQ-MS/MS	Phenomenex Kinetex 2.6u XB-C18 100A (Torrance, CA, USA).	A (0.4% acetic acid in water); B (acetonitrile)	1:5	[87]
Fresh tea leaves, Black tea, Yellow tea	Four hydroxycinnamoylated catechins (HCCs)	UPLC-ESI-MS/MS	An ACQUITY UPLCBEH Shield RP18 column (2.1 × 50 mm, 1.7 μm)	A (formic acid/water, 1/999, mL/mL); B (formic acid/acetonitrile, 1/999, mL/mL)	1:400	[16]
Six types of processed tea and Fresh tea leaves	Seven flavoalkaloids	UHPLC-Q-TOF-MS	An ACQUITY UPLC HSS T3 column (2.1 × 100 mm, 1.8 µm)	A (0.1% formic acid in water); B (acetonitrile)	1:200	[12]

## Data Availability

Not applicable.

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
