# Peer review of "Recent Advances in Analytical Methods for Determination of Polyphenols in Tea: A Comprehensive Review"

_foods, 2022, doi:10.3390/foods11101425_

Round 1

Reviewer 1 Report

The article “Recent Advances in Analytical Methods for Determination of Polyphenols in Tea: A Comprehensive Review” by P. Yin and colleagues reports current methods exploited for detecting and quantifying total and individual polyphenols. The review is interesting although many errors are present and the manuscript can be improved significantly with some modifications.

Line 99 and Figure 2: the basic structure is not that of benzopyridine (containing an aromatic ring with a N) but it is a derivative of benzopyran or chromane. In the central image of Figure 2 it is reported as 4H-chromene (also missing the H in italics) but this derivative should contain a double bond (not represented in the chemical structure and whose position is determined by the 4H), so it must be corrected to “chromane”.

In the main text sometimes it would be more convenient to name the rings in the basic structure as A, B, and C to avoid confusion with the nomenclature.

Finally, texts in Figure 2 are too small and difficult to read.

Figure 3: Top (total tea polyphenols) and bottom (individual tea polyphenols) sections must be made homogeneous in their classification (top: vertical; bottom: horizontal).

Also correct “Folin-Phenol” to “Folin-Ciocalteu”.

Figure 4: no differences can be noted between the 2 “Before” and “After” drawings.

Addition in Section 3 of a table with reported MW and corresponding peaks of the most common polyphenols would be of interest to many researchers and would probably attract many citations.

Minor English and editing errors are present and must be corrected (e.g. “ml” to “mL”, spaces, italics in -N- and -O-, line 300, line 442, etc).

Author Response

Dear reviewer, the point-by-point responses to your comments are in the Word file. Please see the attachment.

Reviewer 2 Report

This submitted review paper provides an overview on analytical methods for determination of total and individual polyphenols in tea and summarizes the recent advance in this field. The approach is comprehensive and analytical, the selected literature is current and adequate, and the methods are well discussed, with a focus on the advantages and disadvantages of each method. Take into consideration addition of below suggestion/comments:

Abstract should improve with more author's observed and concluding remarks.

Line 30, Latin name of plant must be in Italics.

Line 40-41, correct as „5 times more than that of green tea“,  to make sense.

Figure 1. degrees of fermentation are not presented.

Line 60, Which are the four major types of tea polyphenols?

Line 111-114, The content of total catechins in white tea?

Line 116-120, provide citation

Line 129, quercetin is not a good name for the compound because it lists the glycosides. Did you mean quercetin-3-glycoside?

Line 167-173, refer to some citations.

In Table 2, avoid abbreviations or introduce them, for example Infrared scpetroscopy (IR)

Line 182, corrects as „...for the determination of the total tea polyphenols“

For the subsection 3.1. and 3.2. include more examples and references for using these methods.

Line 205-207, unfinished sentence.

Subsection 3.5. Include more examples, or compare with findings of Hidayat et al. (2016), who also used paper based sensor method for determination of tea polyphenols.

Section Simultaneous determination of individual polyphenols cannot be numbered same as section Quantification of total tea polyphenols. Mark with number 4 and further subsections with 4.1. etc.

Line 346-353, Line 544- 564, add citations

In section conclusion and future trends mention rapid electrochemical sensor analysis.

Check and make sure that all journal titles are abbreviated to provide references to be in a consistent format. References 8, 29, 31, 32, 34, 42 full journal titles

Author Response

(The authors gave the same response as above.)

Round 2

Reviewer 1 Report

The revised version of the manuscript is significantly improved and, in my opinion, some minor English (e.g. line 102) and editing errors should be correct.

Author Response

Dear reviewer and editor, please see the attachment.
